# The Redox Potential of the β-^93^-Cysteine Thiol Group in Human Hemoglobin Estimated from In Vitro Oxidant Challenge Experiments

**DOI:** 10.3390/molecules26092528

**Published:** 2021-04-26

**Authors:** Federico Maria Rubino

**Affiliations:** LaTMA Laboratory for Analytical Toxicology and Metabonomics, Department of Health Sciences, Università degli Studi di Milano at “Ospedale San Paolo” v. A. di Rudinì 8, I-20142 Milano, Italy; federico.rubino@unimi.it

**Keywords:** glutathione, glutathione disulfide, glutathionyl-hemoglobin, hemoglobin, hydroperoxide, oxidative stress, red blood cell

## Abstract

Glutathionyl hemoglobin is a minor form of hemoglobin with intriguing properties. The measurement of the redox potential of its reactive β-^93^-Cysteine is useful to improve understanding of the response of erythrocytes to transient and chronic conditions of oxidative stress, where the level of glutathionyl hemoglobin is increased. An independent literature experiment describes the recovery of human erythrocytes exposed to an oxidant burst by measuring glutathione, glutathione disulfide and glutathionyl hemoglobin in a two-hour period. This article calculates a value for the redox potential E_0_ of the β-^93^-Cysteine, considering the erythrocyte as a closed system at equilibrium described by the Nernst equation and using the measurements of the literature experiment. The obtained value of E_0_ of −121 mV at pH 7.4 places hemoglobin as the most oxidizing thiol of the erythrocyte. By using as synthetic indicators of the concentrations the electrochemical potentials of the two main redox pairs in the erythrocytes, those of glutathione–glutathione disulfide and of glutathionyl–hemoglobin, the mechanism of the recovery phase can be hypothesized. Hemoglobin acts as the redox buffer that scavenges oxidized glutathione in the oxidative phase and releases it in the recovery phase, by acting as the substrate of the NAD(P)H-cofactored enzymes.

## 1. Introduction

The red blood cell (RBC), or erythrocyte, is the most abundant single cell type in the human organism and is essential to deliver atmospheric oxygen to the most remote districts of the body. Mature human RBCs are de-nucleated, do not contain most of the cell machinery necessary for protein biosynthesis and thrive on a simplified energetic metabolism of aerobic glycolysis. Due to the large amount of molecular oxygen that they contain and to the presence of ionic iron embedded in the heme group of hemoglobin, the integrity of their molecular structures is at risk of oxidative damage throughout the approx. 120 days of existence of the cells. For this reason, the RBC contains a multi-component anti-oxidant system aimed at quenching the off-products of oxygen activation and their reaction products with the organic molecules and biomolecular structures. Glutathione and hemoglobin are the two most abundant components of a tier of molecular anti-oxidants and enzymes that use the reducing power of NAD(P)H and the metabolic energy of glycolysis to maintain a dynamic equilibrium of intracellular reduced and oxidized chemical species.

Glutathionylated human hemoglobin (HbSSG) is a minor form of hemoglobin that carries a glutathione (γECG) linked through a disulfide (R_1_CS-SCR_2_) bond to the thiol group of a cysteine residue, mainly characterized as the β-^93^C in a single beta-chain of the tetramer. This form of hemoglobin was first highlighted in 1962 [1] and further characterized in 1986 [2,3,4] as a form that inhibits the polymerization of the sickle-cell Hb-S mutated form.

In the tetrameric holo-hemoglobin, glutathionylation occurs on different molecular forms that include met-hemoglobin (that which has one heme-bound iron in the higher-oxidized Fe(III) form rather than in the physiologically prevalent Fe(II) oxidation form [5,6]) and other chemical forms of hemoglobin bound to carbon monoxide [7] and to different forms of nitric oxide [8].

Among several further functional studies, molecular dynamics and native mass spectrometry demonstrate that modification at the crucial β-^93^C position has a profound effect on oxygen affinity. While thioether formation with *p*-benzoquinone, a toxic abundant component of tobacco smoke, decreases the oxygen affinity of hemoglobin [9,10], upon glutathionylation the molecule of tetrameric hemoglobin increases its oxygen affinity [11]. The interaction of hemoglobin with molecular ROS and NOS oxidants generates several irreversible post-translational modifications, such as oxidation methionine residues to the sulfones and sulfoxides [12], of triptophan residues [12], and the nitration of tyrosines [13]. Higher-valence forms of heme-bound iron catalyze the oxidation of protein tyrosines to dityrosines [14,15,16].

Some glutathione-bound forms of holo-hemoglobin can be differentiated with the use of electrophoresis and specifically detected by immuno-staining with glutathione-specific antibodies. The measurement of HbSSG in the RBCs of subjects of population and clinical studies is performed with analytical techniques, such as liquid chromatography with UV detection or coupled to mass spectrometry and stand-alone mass spectrometry with ESI or MALDI ionization. These techniques detect the dissociated alpha- and beta-chains of hemoglobin and the respective covalently modified forms, the most common being *N*-glycated hemoglobin, the biological indicator of mean glucose concentration used in the management of diabetes, and β-glutathionylated hemoglobin.

β-glutathionylated hemoglobin is found at increased levels when hemoglobin or erythrocytes (red blood cells, RBCs) are subject to an excess of oxidants, such as in several human patho-physiological conditions or in ex vivo, in vitro experiments [17].

Several clinical studies [18,19,20,21,22,23,24,25,26,27,28,29,30,31,32,33] report increased levels of HbSSG in etiologically diverse conditions that include: kidney impairment [18,19,20]; degenerative neurological and psychiatric diseases and treatments (Down’s syndrome [21], Friedreich’s ataxia [22], major depressive disorder [23], epilepsy [24]); diabetes [25,26,27,28]; hyperlipidemia [26]; obesity [29]; iron deficiency anemia [30]; the consequences of tobacco smoking [29]; and the exposure of petrochemical workers to butadiene, a carcinogenic airborne toxicant [31]. Some patients undergoing surgery with high-oxygen ventilation showed very high levels of HbSSG [32]. A transient rise-and-fall phenomenon has been observed in some patients who underwent no-bypass carotid surgery to remove atheromatic plaque [33].

Its physiological formation likely involves complex and still questioned pathways [34] to which contribute the endogenous ROS naturally generated in the RBCs, mostly hydrogen peroxide; a number of different forms of hemoglobin that include different formal valence of the heme iron atom (methemoglobin) and bound gaseous molecules (O_2_, CO [7], NO [8]); the intra-erythrocyte pool of soluble thiols, several enzymes of RBCs in the membrane-bound and soluble cytoplasmatic forms [35,36]; and some crucial signaling mediators, such as carbon monoxide [7] and the chemical forms of nitric oxide, the flux of nutrients, chiefly glucose, from the extra-cellular medium to the RBCs.

A wealth of research addresses individual pathways, but a comprehensive understanding of the role of HbSSG in the RBC physiology and physio-pathology is still far from being understood [37,38]. Consequently, the measurement of glutathionyl-hemoglobin as a clinically relevant bio-marker of oxidative stress has not been exploited, and glutathione and glutathione disulfide levels in RBCs are only measured in the context of research.

Concentration-derived redox potentials as synthetic indicators to describe the balance between oxidant challenge strength and the efficiency of cellular coping mechanisms are still of limited use [39]. Fastidious measurement is necessary to obtain meaningful results [40,41] and there is a pending strong debate on the objection that a purely thermodynamic, or electrochemical, description would not consider the necessity of the intervention of enzymes for the redox/trans-thiolation reactions to proceed at a velocity compatible with biological processes [42,43,44,45,46].

The availability of values for the redox potentials of more couples, i.e., of more nodes in the circuitry for biological redox signaling and control [38], may improve the quantitative description of the intracellular redox state [34,37] in the dynamic behavior of cells subject to physio-pathological events.

To contribute to this process, this article reports the calculation of the redox potential of the thiol group of human hemoglobin where glutathionylation mostly occurs. The author of this article re-elaborates literature results that Colombo et al. independently obtained and previously published in 2010 [47] and performs on these his original calculation.

## 2. Results

Colombo et al. [47] published in 2010 results of an experiment where ex vivo, in vitro red blood cells (RBCs) were diluted into glucose-containing buffered saline and treated with an organic free radical generator, *t*-butyl-hydroperoxide, that is able to freely penetrate cell membranes [48,49]. The complete experiment was performed on RBCs taken both from humans and from rats. Here, only the results of the experiment on human RBCs are considered.

### 2.1. Ex Vivo, In Vitro Oxidative Challenge of Red Blood Cells by Hydroperoxides

Once inside, the hydroperoxide oxidizes free thiols to the respective sulfinic acids (reaction Equation (1)) which then react with excess thiols to produce the disulfides (reaction Equation (2)) [48].
*t*Bu-OOH + R_1_-SH → *t*Bu-OH + R_1_-S-OH(1)
R_1_-S-OH + R_2_-SH → R_1_-S-S-R_2_ + H_2_O(2)
(R_1_ = GSH, R_2_ = GSH or HbSH)

The intermediacy of sulfenic acid is demonstrated both for the small-molecule soluble thiols and for proteins. In particular, the same research group had shown that the initially formed cysteine sulfenic intermediates rearrange into the corresponding thio-aldehyde, and through subsequent hydrolysis the C-3 methylene of cysteine transforms into an aldehyde group, which at last can react with a standard carbonyl reactive, 2,4-dinitrophenyl-hydrazine, to generate the expected hydrazone [50]. In addition, they reported the direct observation of β-^93^C-Hb sulfenic acid in the infusion ESI spectrum of human hemoglobin oxidized in vitro with hydrogen peroxide [51].

In the experiment, timed samples of the mixture were taken over approx. two hours after the addition of the oxidant. The authors measured the concentration of the three main products, namely glutathione in its “reduced” thiol form (GSH), in its “oxidized” disulfide form (GSSG) and of glutathione bound as its mixed disulfide with RBC proteins, which mostly consists of the hemoglobin mixed disulfide (HbSSG). The time course of the concentrations of glutathione, glutathione disulfide, glutathionyl-hemoglobin and total glutathione in the experiment, recalculated from the data in the original article, is summarized in the graph of Figure 1.

The treatment quickly caused a sharp decrease in the concentration of GSH and a stoichiometrically equivalent formation of GSSG and HbSSG. Within two hours, the metabolically viable RBCs restored GSH, GSSG and HbSSG to the respective pre-treatment levels.

The time curves of a decrease in glutathione disulfide and glutathionyl-hemoglobin and of an increase in glutathione show a close quantitative correspondence in the restore phase between the production of glutathione and the disappearance of glutathione disulfide and glutathionyl-hemoglobin. The appreciably linear trend from 40 to 60 min can be approximated to a first-order kinetics. In particular, the increase in GSH evaluated through the slope of the linear trend (k = 0.033 min^−1^) closely parallels the decrease in HbSSG (k = −0.032 min^−1^), while the slope of GSSG disappearance is much steeper (k = −0.085 min^−1^), as shown in Figure 1.

This phenomenon, coupled with the observation of a constant size of the total glutathione pool during the whole experiment (the sum of the concentrations of GSH, GSSH and HbSSG is 1.26 ± 0.02 mM), encourages considering this system as closed with respect to the interplay of the redox reactions that involve glutathione. In addition, the treatment involves a single burst of the oxidant and a complete recovery of the pre-treatment levels of the glutathione pool. These combined criteria encourage considering the involved redox systems as reversible during the experiment.

### 2.2. Redox Equilibria of the Bio-Thiols and Measurement of the Standard Electrochemical Potential (E_0_)

The two reactions that restore “reduced” glutathione level in the oxidatively challenged RBCs can be interpreted as the semi-element of complete electrochemical cells (reaction reaction Equations (3) and (4)):GSSG + 2H^+^ + 2e^−^ ⇆ 2 GSH(3)
Hb-^93^C-SH-SG + 2H^+^ + 2e^−^ ⇆ Hb-^93^C-SH + GSH(4)
that are coupled to the reducing activity of the viable RBC through its several biochemical pathways, enzymes, cofactors and use as a nutrient and source of reducing power the glucose supplied in the incubation medium.

In addition, the two reactions share glutathione as the common participant and can mechanistically couple to each other through the known chemical mechanism of thiolate interchange (reaction Equation (5)) that operates in the physiological cellular phenomena of glutathione-controlled redox regulation.
R_1_S^−^ + R_2_S−SR ⇆ R_1_S−SR_2_ + ^−^SR(5)

The concentration(s) and chemical characteristics of each thiol in the biochemical grid of glutathione biosynthesis, catabolism and redox equilibria can be summarized, for each involved thiol, and characterized by an individual “R-group”, through the value of the concentration-dependent redox potential E_h_, calculated according to the general Nernst equation (Equation (6)):E_h_ = E_0_(RS) + RT/nF ln [RSSR]/[RSH]^2^(6)

The values of E_0_ of several components of the metabolic grid of glutathione and of a few related xenobiotics and drugs have been measured by exploiting the analytical measurement of the concentrations of the “reduced” (free thiol) and “oxidized” (disulfide) form of a measurand thiol (RS) and of a reference thiol (RefS) in a chemical system that is brought at an equilibrium, according to the reaction Equation (7):RefS-SRef + 2 RSH ⇆ 2 RefSH + RSSR(7)
and the corresponding quantitative relationship of Equation (8):E_0_(RS) + RT/nF ln([OX_RS_]/[RED_RS_]^2^) =             E_0_(RefS) + RT/nF ln([OX_RefS_]/[RED_Ref_]^2^)(8)
that uses the measured equilibrium concentrations of the Oxidized (disulfide) and REDuced (thiol) forms of each compound and the value of the E_0_ of the reference thiol to calculate the E_0_ of the measurand thiol.

Different physico-chemical methods, such as chromatographic separation and Nuclear Magnetic Resonance spectroscopy, have been used to measure the concentrations of the thiol and disulfide forms of two competing thiols. The chemical mechanism of the trans-thiolation reaction (disulfide exchange and reduction of a more oxidizing thiol disulfide by a more reducing thiol) depends on the thiolate form of the reducing thiol as the active species. This mechanism is also confirmed by the observation that more reducing small-molecule thiols are, in general, also more acidic [52].

Physiologically, the metabolic response to oxidative challenge involves several enzymes that contain disulfide and thiol active sites. These enzymes accelerate the equilibration of thiol–disulfide mixtures. For this reason, a catalytic amount of an enzyme such as glutathione reductase (EC 1.8.1.7; at: https://www.brenda-enzymes.org/enzyme.php?ecno=1.8.1.7, accessed on 23 April 2021) was added to the reaction mixtures on which the measurement of thiol–disulfide equilibria in purely bio-mimetic systems was performed. This addition does not modify the relative equilibrium concentrations of the chemical forms but only accelerates the reaction [53].

This overall picture of the reversible nature of thiol–disulfide exchange in biomimetic systems encourages the use of the Colombo 2010 experiment for calculating a redox potential, E_0_(HbS), for the thiol group of hemoglobin that is involved in the formation of glutathionyl-hemoglobin. The knowledge of this value allows comparing the position of hemoglobin’s β-^93^C thiol group within the series of bio-thiols of the red blood cell and to better appreciate the interplay of reactions that take place in the erythrocyte under several physiological and pathological conditions.

### 2.3. Calculation of the Redox Potential of the Thiol Group of Hemoglobin’s Beta-93 Cysteine

Due to the unlikely formation of a “dimeric hemoglobin” form with two hemoglobin tetramers held together by an inter-chain disulfide bond, the oxidized form of the tetramer considered in the electrochemical semi-cell is considered as involving an intermediate sulfenic acid. This choice is in accordance with the results of in vitro mechanistic experiments where the intermediate species was generated by the reaction of hemoglobin with hydrogen peroxide, identified by mass spectrometry, and reacted with glutathione to generate HbSSG, according to the reaction of Equation (9) [51]:Hb-^93^C-SH + H_2_O_2_ ⇆ Hb-^93^C-S-O^−^ + H_2_O + H^+^(9)

The sulfenic acid intermediate is transient and reacts quickly with the soluble, low-mass thiols present in the RBC cytoplasm, the most abundant being glutathione, to generate the corresponding hemoglobin mixed disulfide, which is the measured species, according to the reaction of Equation (10):Hb-β-^93^C-S-O^−^ + GSH + H^+^ ⇆ Hb-β-^93^C-S-SG- + H_2_O(10)

Due to the relatively low concentration of the mixed disulfide in comparison to that of glutathione (hardly exceeding 5% of the glutathione pool and 2% of the hemoglobin pool, immediately after oxidant addition), essentially all the formed cysteine sulfenic intermediate is converted into the measured form, glutathionyl-hemoglobin. Therefore, the concentration of HbSSG is a suitable proxy for that of the sulfinic acid intermediate.

The overall process is thus described, in the electrochemical standard notation, according to the electrochemical chain of Equation (11), as two half-cell elements, one being the glutathione disulfide–glutathione pair, the other the hemoglobin β-^93^C thiol group in its reduced (thiol(ate) C-SH or C-S-) and oxidized form (sulfinic acid, C-S-OH or its deprotonated form).
GSSG/GSH//Hb-^93^C-S-SG/ Hb-^93^C-SH(11)

To perform this calculation, it is assumed that at all sampling times during the experiment the electrochemical potential is the same for the two semi-elements, according to the respective concentrations of the “oxidized” and “reduced” component. The Nernst equation is thus formulated as Equation (12):E_eq_ = E_0_(GS) + RT/nF ln [GSSG]/[GSH]^2^ =            E_0_(Hb) + RT/nF ln [HbSSG]/[HbSH](12)

Thus, by knowing the concentrations (proxies of the activity) of the four involved chemical species and using as reference the value of the standard potential E_0_ for the glutathione redox couple, it is possible to calculate the corresponding standard potential for the oxidation of the β-^93^C thiol group of cysteine in hemoglobin in the same conditions. Since the E_0_ of the disulfide-thiol(ate) system depends on the pH of the solution, the generally used value for biochemical studies is −264 mV at the physiological pH 7.4. This value was calculated from the E_0_ value −240 mV at pH 7.0 and considering a pH effect of −59 mV/pH unit [34,35].

The tabulated concentrations and calculations are reported in Table A1 of the Appendix A. The calculation has been performed in two tiers of improving approximation.

#### 2.3.1. First-Tier Calculation

An initial approximation of the E_0_(HbS) has been calculated by simply rearranging the Nernst redox equation (Equation (12)) for equilibrium conditions at all times, as in Equation (13):E_0_(Hb) = E_0_(GS) + RT/nF ln [GSSG]/[GSH]^2^ − RT/nF ln [HbSSG]/[HbSH](13)
and calculating, according to the concentrations measured in the samples retrieved at each time point during the post-challenge recovery phase of the experiment, the individual values of E_0_. The calculated E_0_ values are not constant over time (see Section 2.4 below) and feature a sigmoidal curve with a mean value of −121 mV, the first derivative of which peaks between 40 and 60 min, where E_0_ has a mean value of −127 mV (Appendix B, Figure A2).

#### 2.3.2. Second-Tier Calculation

To improve the approximation, the time curve of HbSSG, calculated as percent of total hemoglobin, is evaluated with the Nernst equation from the measured concentrations of HbSSG and of HbSH, changing the value of E_0_ between −130 and −115 mV in 1-mV steps to minimize the least-squares difference between calculated and measured HbSSG levels. The value of E_0_ that affords the best approximate of the measured concentration of HbSSG in the experimental system with the least-squares method is −121 mV (Appendix B, Figure A3).

To confirm the adequacy of the value of E_0_ calculated with the least-squares approximation, the levels of HbSSG (as percent of free-Hb) were calculated employing values of E_0_ from −130 to −115 mV (those used to build the least-squares minimization curve of Figure A3). The concentration ratio-dependent term of the Nernst equation used the corresponding timed concentrations of GSSG and GSH and the corresponding values of E_h_.

The graph of Figure 2 shows some results as the time profile of HbSSG% during the experiment.

The levels of HbSSG (as %) calculated using the best-estimate E_0_(HbS) value of −121 mV show an excellent match of calculated (dashed line) to measured values (closed triangles) over the entire range of the experiment. On the contrary, higher and lower values differing by as little as ±1 mV generate analytically appreciable differences from actual measurements.

Therefore, an E_0_ value of −121 mV at the physiological pH 7.4 is deemed a reliable estimate and can be used in the further calculations that aim at understanding the role of glutathionylated hemoglobin in oxidant-challenged RBCs.

### 2.4. Coping with Oxidative Stress and Redox Potentials of the Thiols in the Red Blood Cells

#### 2.4.1. Placement of Glutathionyl-Hemoglobin among the Redox Bio-Thiols

The evaluated E_0_ value of −121 mV at pH 7.4 places the β-^93^C thiol group of cysteine in hemoglobin as the least reducing among those of the bio-thiols of the red blood cell (Table 1), or as that less prone to oxidation. 

This large difference of the E_0_ means that the glutathionylation of hemoglobin only takes place when the soluble pool of thiols is close to being exhausted by too high a level of intracellular oxidants, or from the impairment of recycling glutathione disulfide, or by an insufficient size of the intracellular glutathione pool.

The concentration of hemoglobin inside the human RBC, at 5.5 mM, is ten to fivefold higher than that of the total glutathione pool (a total 1.2 mM concentration is measured in the examined experiment). Therefore, even a small level of HbSSG (measured as percent, with reference to the hemoglobin pool) corresponds to a fivefold higher fraction, when referred to the glutathione pool. 

In addition, the redox curves of glutathione and hemoglobin in the RBC (plotted as E_h_ of the half-cell as a function of the oxidized fraction of each thiol, see Appendix C, Figure A4) are very different, due both to the large difference in the size of the two intracellular thiol pools and to the respective E_0_ values.

A major contribution to the different shape of the two curves is that, for the glutathione disulfide/glutathione redox couple, the concentration-dependent addend of the Nernst equation depends on the square of the concentration of GSH, but only on the concentration of GSSH. Thus, it also depends on the size of the intra-erythrocyte pool of glutathione, which is a genetically determined trait [55]. On the contrary, for hemoglobin it depends only on the ratio of the oxidized and reduced form, but not on the absolute concentration of hemoglobin. In addition, hemoglobin concentration in the individual RBC is in most cases constant, and most commonly anemia occurs through a reduced number of RBCs in blood (a low value for the hematocrit), rather than of a low concentration of hemoglobin inside the individual RBCs.

#### 2.4.2. Towards a Mechanistic Understanding of the RBC Response to Oxidative Burst

During the experiment, the total oxidized fraction (χ_ox_%) of the total glutathione pool varies between a pre-challenge fraction of approx. 2% to a maximum of approx. 95% immediately after the addition of the oxidant, and decreases to reach the pre-challenge value at the end of the two-hour experiment. At the different times, this “oxidized glutathione pool” is present both as soluble disulfide, GSSG and as hemoglobin-bound glutathione (that bound to other soluble and membrane proteins is considered as of a negligible amount in this first-tier model), in variable proportions. 

The recovery of the RBC after the oxidative burst in the examined, simplified biological system of the Colombo experiment can be examined with multiple metrics. The starting point is verifying mass balance and the rate of replenishment of the reduced glutathione pool from both its sources as the disulfide-bound species, glutathione disulfide and glutathionyl-hemoglobin. This preliminary step is displayed in the histogram of Appendix C, Figure A4, which shows the almost matching correspondence of glutathione disulfide generated during the oxidative burst with that produced over time from glutathione disulfide and from glutathionyl hemoglobin.

Complementary suitable metrics for this calculation are the “branching ratio” (χ%) for the formation of either GSSG or HbSSG, calculated according to the Equation (14) and displayed in the graph of Figure 3: χ_ox_% = χ_GS_% + χ_HbS_%(14)

The mechanism for the recovery of the RBC after the oxidative burst in the examined, simplified biological system of the Colombo experiment can be described by referring to the E_h_ curves of both redox couples, rather than to the concentrations and their ratios [39]. The measured concentrations of GSH, GSSG, HbSSG and the values of E_0_ for glutathione (−264 mV) and for hemoglobin (−121 mV) allow the calculation of the concentration-dependent E_h_ electrochemical potentials of the two coupled redox pairs during the experiment. Their time evolution is reported in the graph of Figure 4.

As apparent, during the recovery phase after the oxidative burst, the two time curves show a very different profile, with the E_h_ of the glutathione pair at a less negative value earlier in the experiment, and a reversal of the potential difference at the end, where the potential of the glutathione pair is lower than that of the hemoglobin pair. The two-electron transfer occurs as a thiol–disulfide exchange from the more negative, more reducing element of the formal electrochemical cell towards the less negative, more oxidizing one.

In the earlier phase, before the intersection point, hemoglobin in its thiol, reduced form is the reducing element of the electrochemical cell, according to reaction Equation (15):HbSH + GS-SG ⇆ HbS-SG + GSH(15)

In the later phase, after the intersection point, electron transfer occurs from the now more reducing element of the cell, glutathione in the thiol form, towards the now more oxidized element, the disulfide glutathionyl-hemoglobin, according to reaction Equation (16):GSH + HbS-SG ⇆ GS-SG + HbSH(16)

The crossing of the curves occurs around 45 min from the start of the experiment, or 25 min from start of the recovery phase. At this time, the calculated value of the potential is approx. −168 mV. At the crossing point, the two reactions occur at the same velocity and the corresponding concentrations of the four species can be calculated from Equation (17):E_eq_ = E_0_(GS) + RT/nF ln [GSSG]/[GSH]^2^ =            E_0_(Hb) + RT/nF ln [HbSSG]/[HbSH](17)
and from the corresponding matter balance.

This experiment does not give, in itself, any information on the flux of reduction that recovers glutathione thiol after the oxidative burst. A complementary way to envision the biological pathways of RBC recovery from the oxidant burst can be achieved by plotting the respective values of the E_h_ potentials of the two redox couples at the different times of the experiment, as shown in Figure 5. To express the concentrations of the thiols and disulfides as redox potentials has an “added value” in simplifying the description of the phenomenon. This is because the calculated E_h_ values also include the E_0_ standard potential as an indication of which of the involved thiols is the more strongly reducing one. The concentration-dependent term of the Nernst potential describes how the concentration ratio influences the “per se”, structure-dependent, stronger or weaker oxidizing character of each thiol. In the case of glutathione and its disulfide, the concentration-dependent Nernstian term depends *both* on the absolute concentration of the (soluble) glutathione pool *and* on the concentration ratio of the thiol and disulfide forms. Electrochemical potentials are thus more flexible synthetic indicators than measured concentrations, because more information is embedded in them.

In this depiction, the dashed diagonal line represents an (entirely formal) equilibrium situation where the two couples have the same potential and the electromotive force (EMF, mV) of the system along the line is zero, so that no net electrochemical energy is generated or absorbed. In the frame of reference adopted for the axes of the graph, “spontaneous” electron flow proceeds from the lower (more negative) potential to the higher (less negative). At each time point, the EMF of the RBC is represented by the orthogonal or Euclidean distance of each point from the equilibrium line. An example is shown in the Figure 5 for the 20 min time point, at the beginning of the recovery phase.

This description of the system employs the two main electrochemical couples of the RBC. At the beginning of the experiment, before oxidative burst (t = 0 min. closed diamond), the RBC system is in a reduced state, at approx. −240 mV. The two redox couples are close to the equilibrium, with only a few mV EMF of the HbSH couple keeping the bulk (more than 98%) of the glutathione pool in the reduced form, and with more than 95% of the oxidized pool present as glutathione disulfide.

In this simplified experiment, the oxidative event (red dashed arrow) is essentially instantaneous and the recovery phase starts immediately after the oxidant has been completely consumed.

At the first time point (t = 10 min after the oxidative burst; red dashed arrow), the oxidation of the thiol pool (GSH and HbSH) has taken place and has produced the bulk of GSSG and HbSSG. The increased concentrations of both raise the E_h_ potential of each couple to less negative values, distant from the equilibrium line, with the glutathione pair being displaced from its pre-challenge value more (from −235 mV to −107 mV) than that of hemoglobin (from −239 mV to −176 mV).

As the time course of the concentrations of GSH, GSSG and HbSSG of Figure 1 describes, the RBCs return essentially to their pre-challenge status within the two-hour experiment. In principle, the system may follow any path to resume the pre-challenge concentrations of the involved thiols. 

The RBC accomplishes this process by using the metabolic reducing power of NAD(P)H and different reducing enzymes both in their membrane-bound and soluble cytoplasmic form. Recovery entails the reduction of both glutathione disulfide and glutathionyl hemoglobin to glutathione in its thiol form. Literature studies demonstrate that direct thiol–disulfide interchange is a very slow and inefficient process. In particular, an experiment with radioactive ^35^S-glutathione-HbS-^35^SG demonstrates that thiol glutathione reacts only slowly with HbSSG to generate radiolabeled glutathione disulfide; however, the reaction with NADPH catalyzed by glutathione reductase liberates radiolabeled GSH with much faster kinetics [56].

The pathway that the RBC follows towards recovery can be described by following over time the successive values of the E_h_ of each redox pair in the graph of Figure 5. The RBC does not follow back the direct pathway of the oxidative burst. On the contrary, the plot of the potentials allows the identification of two time phases. At first (t = 10 to 60 min, first highlighted compartment, crossover time at approx. 46 min, as shown in Figure 4), the E_h_ of the glutathione pair recovers, from the minimum of −107 mV to −168 mV at 46 min and −214 mV at 60 min, while that of hemoglobin only recovers from −176 mV to −193 mV. In the second phase of the recovery phenomenon, the E_h_ of the hemoglobin redox pair recovers most of its value, from −168 mV at approx. 46 min (Figure 5) to −193 mV at 60 min and to an average of −225 mV between 100 and 120 min.

The histogram of Figure A5 displays the calculated regeneration rates of glutathione thiol from both glutathione disulfide and glutathionyl-hemoglobin. Each rate is calculated as the difference in the concentration of the disulfide form between successive time points, which are apart by ten minutes. The graph shows that, while the regeneration rate of glutathione from glutathione disulfide shows large variations, in a bell-shaped time profile, from one observation time to the following, glutathione re-generation from glutathionyl hemoglobin does not feature large differences, especially in the earlier phase of the recovery.

## 3. Discussion

Independent re-evaluation of the results of the Colombo 2010 experiment allows the assignment of a value of −121 mV at the physiological 7.4 pH to the E_0_ electrochemical reduction potential of the β-^93^-Cysteine residue of human hemoglobin. This cysteine residue combines with glutathione to yield the main form of glutathionylated hemoglobin. The obtained value corresponds to a very oxidizing thiol, when compared to glutathione, which is itself the most oxidizing compound among the physiologically occurring non-protein soluble thiols within the red blood cell.

One important biochemical scenario is the mechanism whereby the RBC exposed to a burst of a radical-generating substance copes with the resulting oxidative stress and recovers to the previous condition. The overall occurring phenomenon can be summarized in the scheme of Figure 6. 

As a first-tier interpretation of the general phenomenon that occurs inside oxidatively challenged RBCs, HbSH acts as a buffer scavenger of the initial, transient form of oxidized glutathione (glutathione sulfinic acid, GS-OH) to yield the mixed disulfide. In this way, there is a 50% sparing of soluble glutathione in the reduced form due to the reaction of glutathione sulfinic acid with the large, five-to-tenfold excess of hemoglobin in the RBC, a phenomenon that doubles, on a stoichiometric basis, the oxidant-coping capacity of GSH in the RBC.

Thus, in the experiment, the oxidant burst generates glutathione sulfenic acid (GS-OH) as the first-tier species. The sulfenic acid reacts with any present nucleophile, mostly with thiol(ate) groups from protein- and non-protein cysteines, to yield the corresponding disulfides. The two most abundant thiols present inside the RBC are hemoglobin (approx. 5.5 mM) and glutathione, at concentrations (as the total pool of the thiol and disulfide form) that vary from 0.5 to 4 mM [55]; in the considered experiment, its total concentration is 1.26 mM.

Another formally symmetrical reaction pathway would generate the sulfenic acid intermediate from the the β-^93^-Cysteine residue of hemoglobin, which would then react with glutathione to yield HbSSG [51] (reaction Equation (18)).
HbS-OH + GSSG → HbSSG + GSH(18)

In the examined experiment, the alternative pathways cannot be discriminated, since the respective sulfenic acid intermediates were not identified or measured.

## 4. Materials and Methods

As stated in the introduction, the experimental data on which this article is construed are reported in the cited publication [47] and the author of this article had no role in it.

The author of this study retrieved the results by manual measurement of the graphs in the original article and back-calculating them into the units suitable for the calculations that constitute the bulk of this article. All calculations in this study were performed by the author in original Microsoft Excel spreadsheets.

Briefly, the authors of the original study obtained human blood from volunteers and prepared glucose-saline washed red blood cells at a measured concentration of 5.2 × 10^−3^ mol/L (measured as hemoglobin concentration) for the oxidant challenge experiment. The oxidant challenge experiment involved the addition to the suspension of RBCs of the membrane-crossing *t*-butyl-hydroperoxide, at a concentration sufficient to oxidize the RBC glutathione pool. Samples of the RBCs were withdrawn at timed intervals within 2 h to measure the concentration of three chemical species: glutathione thiol (GSH), glutathione disulfide (GSSG), glutathionylated hemoglobin (HbSSG). The authors employed liquid chromatography to measure GSH and GSSG, and HbSSG was measured with a glutathione antibody following electrophoresis of the RBC proteins. The concentrations of all measured compounds were expressed in mol/L.

## 5. Conclusions

This study exploits published measurements performed by an independent laboratory to calculate the redox potential of the thiol group of ^93^Cysteine in the β-chain of tetrameric human hemoglobin. The applied calculation method affords a first-tier value of the electrochemical potential, the accuracy of which is deemed sufficient to investigate the behavior of red blood cells exposed to radical-generating chemicals or toxic electrophiles and the individual steps leading to oxidative stress.

Scenarios of interest feature both transient and chronic exposure to conditions that generate oxidative stress. Transient oxidative bursts followed by fast recovery occur in real life situations, such as that observed in cerebral vascular surgery [33] and in the administration of some alkylating pharmaceutical drugs in the treatment of malignancies (unpublished data from this laboratory). Chronic rather than transient exposure to chemical oxidants occurs in diseases or conditions that impair the functioning of the enzymes involved in energy metabolism and in redox homeostasis. Normal, healthy subjects as well may face the consequences of exposure to an overload of oxidants, as the consequence of noxious lifestyles, such as tobacco smoking [29] and unhealthy dietary lifestyles [29]. Working under hazardous exposure to butadiene in a petrochemical plant [30] generates a high level of HbSSG in the workers.

Red blood cells that circulate in the lung are in a condition of high oxygen content, with a fully saturated hemoglobin and a gas-saturated intracellular fluid. Several volatile organic compounds that contaminate ambient air are strongly lipophilic and easily cross the RBC membrane. Although the RBC does not contain a specific biotransformation machinery, such as that which is based on the C-H activating P450 cytochromes, it is long demonstrated that oxy-hemoglobin is per se able to perform activation reactions on suitable organic substrates. One such example is the epoxidation of styrene to its strongly electrophilic and thiol-reactive styrene oxide [57,58,59], independently from systemic liver biotransformation.

This study emphasizes the value of the ex vivo, in vitro experimental model of isolated red blood cells to investigate the mechanism(s) of oxidative stress based on the perturbation of the redox equilibria of the cysteine thiol-based cellular machinery with different reagents and toxicants. The availability of a value for the E_0_ of a further bio-thiol, the reactive cysteine of hemoglobin, may incorporate an additional node in the circuitry for biological redox signaling and control [43] of red blood cells subject to physio-pathological events, with the use of redox potentials as physically meaningful synthetic indicators of oxidative stress.

## Figures and Tables

**Figure 1 molecules-26-02528-f001:**
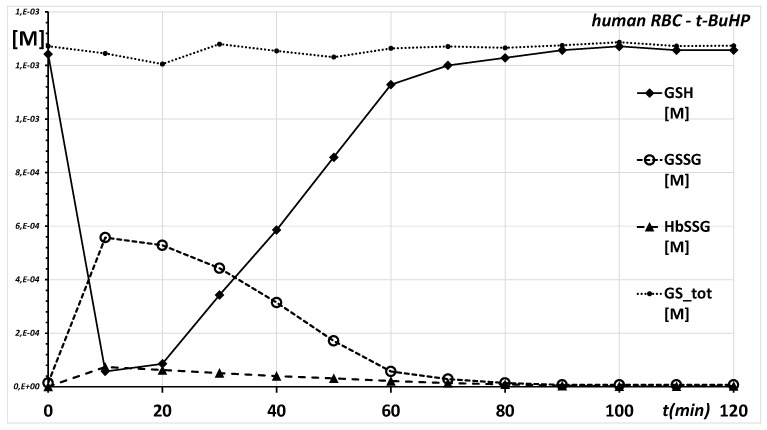
Time course of the concentrations of glutathione (GSH, ♦), glutathione disulfide (GSSG, ◯), glutathionyl-hemoglobin (HbSSG, ▲) and total glutathione (sum of the preceding, ●) in the experiment performed by Colombo et al. [47]. Data have been retrieved, recalculated and replotted from the figures in the original article. Recalculated raw data are reported in tabular form in Appendix A.

**Figure 2 molecules-26-02528-f002:**
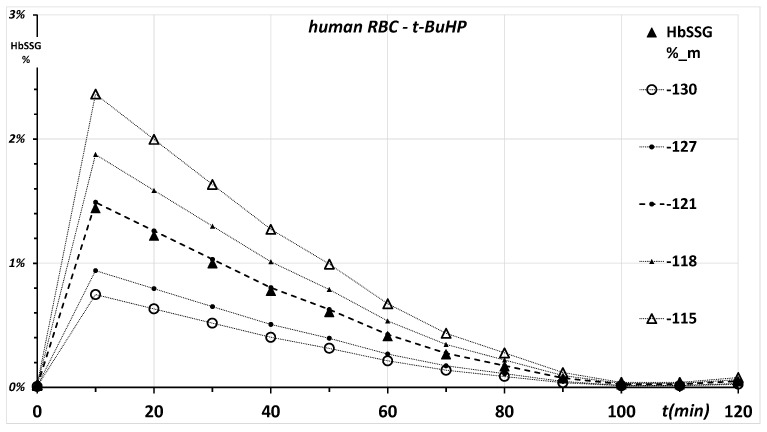
Equilibrium concentrations of glutathionyl-hemoglobin (HbSSG%) calculated according to the Nernst equation. The E_h_ redox potentials correspond to those of the GSSG/GSH redox couple calculated according to the concentrations of GSH and GSSG measured during the experiment. Different estimates of E_0_ for the HbS/HbSH couple are used to derive the concentration ratio dependent term of the Nernst equation for HbSSG/HbSH. For clarity, only two values for higher and two for lower E_0_ of the HbS/HbSH couple are reported.

**Figure 3 molecules-26-02528-f003:**
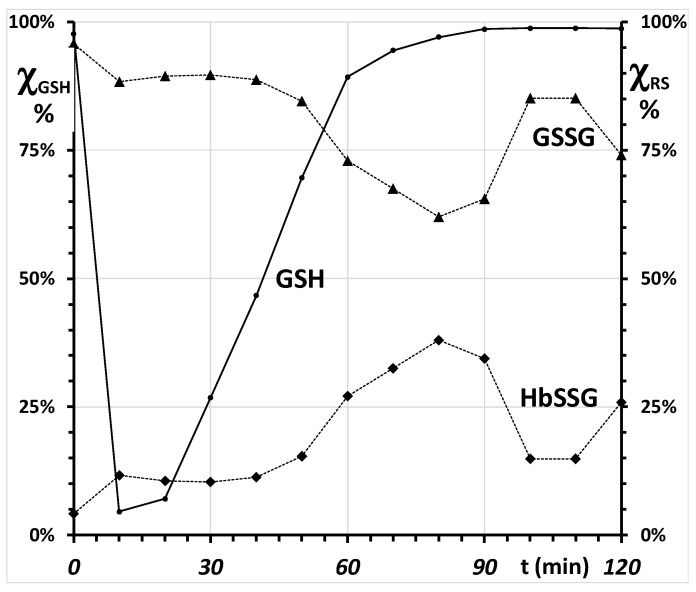
Partitioning of the glutathione pool during the RBC oxidative challenge experiment, as mole fraction (χ%) of each thiol. Mole fraction of reduced glutathione GSH (left); (right) mole fraction of each disulfide form of oxidized glutathione: GSSG (▲) and HbSSG (◆).

**Figure 4 molecules-26-02528-f004:**
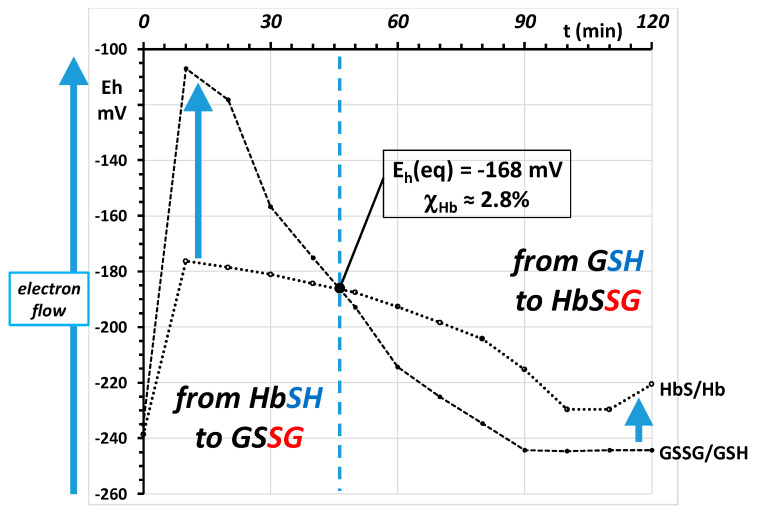
Electrochemical potential (E_h_) of the GSSG/GSH and HbS/HbSH redox pairs during the oxidative burst and recovery experiment. The E_h_ values are calculated on the basis of the measured concentrations of GSH, GSSG, HbSSG and of the values of E_0_ for glutathione (264 mV, literature data) and for hemoglobin (−121 mV, calculated above).

**Figure 5 molecules-26-02528-f005:**
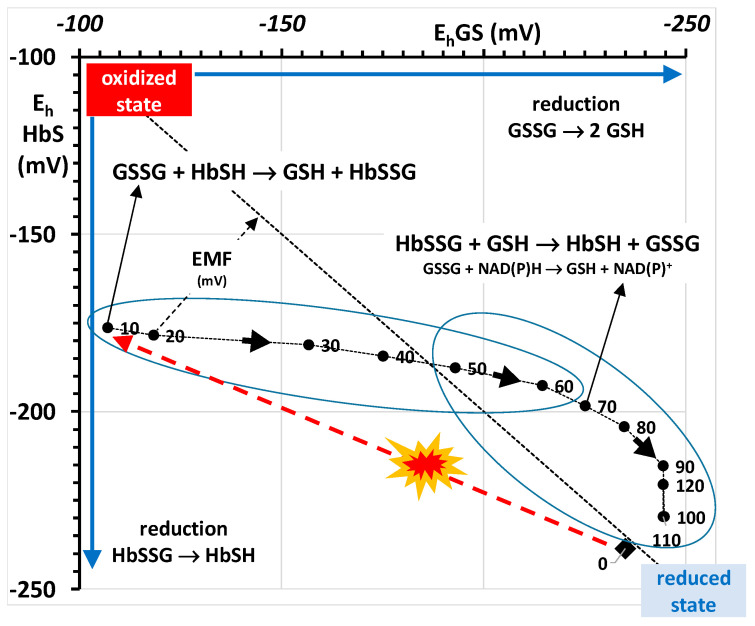
Plot of the respective values of the E_h_ potentials of the two redox couples at the different times during the RBC oxidative challenge experiment. Shown are the values of the E_h_ at the respective times (closed circles, time in min, numbers close to the closed circles, time flow identified by the linking line and the closed arrows).

**Figure 6 molecules-26-02528-f006:**
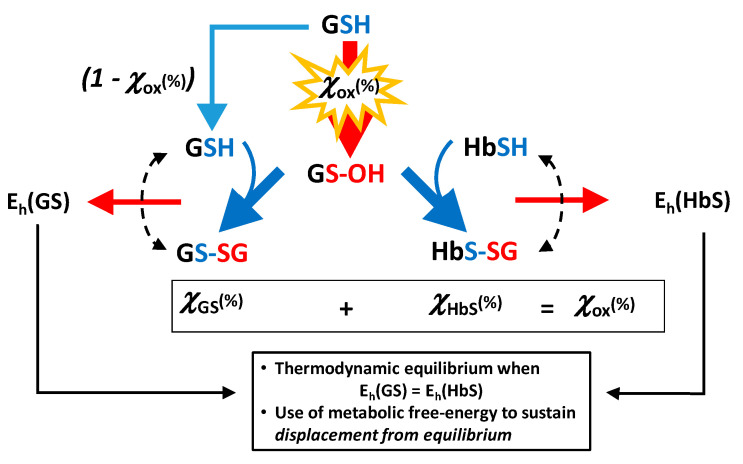
Partial scheme of the formation of oxidized glutathione and glutathionyl-hemoglobin in oxidatively stressed red blood cells.

**Table 1 molecules-26-02528-t001:** Values of the electrochemical reduction potential of the metabolic intermediates of the biosynthesis and catabolism of glutathione and of some compounds of pharmaceutical interest.

Thiol	E_0′_ ^1^	E_0′_ ^2^	E_0_(7.4) ^3^	Ref.
hemoglobin (human)	-	-	−121	This work
glutathione	−205	−205 ± 3	−264	[54]
homocysteine	−196	−218 ± 3	-	[54]
cysteine	−247	−246 ± 3	-	[54]
γGluCySH	-	−265 ± 3	-	[54]
penicillamine	−267	−266 ± 3	-	[54]
*N*-acetylcysteine	-	−268 ± 3	-	[54]
HSCyGly	-	−272 ± 3	-	[54]
cysteine methyl ester	-	−282 ± 3	-	[54]
*N*-acetylpenicillamine	-	−295 ± 4	-	[54]
cysteamine	−203	−372 ± 7	-	[54]

Notes. ^1^ electrochemical reduction potential at standard conditions (mV, pH 1, 1M); ^2^ more electrochemical reduction potentials, measured in the gas-phase by tandem mass spectrometry [54]; ^3^ reduction potentials, calculated at pH 7.4, by considering for glutathione a pH effect of −59 mV/pH unit [39].

## Data Availability

Calculation spreadsheets are available from the author upon reasonable request.

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
