# Peer review of "The Redox Potential of the β-93-Cysteine Thiol Group in Human Hemoglobin Estimated from In Vitro Oxidant Challenge Experiments"

_molecules, 2021, doi:10.3390/molecules26092528_

Round 1
Reviewer 1 Report
The reviewed paper is dedicated to very interesting biochemical field. This is the such kind of hemoglobin modification as glutathionylation, connected with the oxidation/reduction process of cysteine residues. The calculation of redox potential of cysteine thiol group provides new data to explain this phenomenon. These results are important both for a fundamental point of view and give an explanation for some biomedical observations.
But it is possible to make some minor comments, that would be good to correct..
The manuscript contains a well-written review in the section 1 "Introduction". It describes in detail the effect of oxidative stress on cyseine residues in hemoglobin. Indeed, these residues are the most reactive and susceptible to oxidizing agents. However, there are some other effects of the oxidation action on Hb, e.g. the formation of dithyrosines or the oxidation of triptophan residues. It may be useful to briefly mention this, so that the reader does not think that only cysteine residues are modified.
When start reading the section 2 "Resuls", you find yourself thinking, where the results of the author of the article are, and where the literary data is. And only in the section 4 "Materials and Methods" one can read that : "The experimental data on which the article construres are reported in the cited publication [42] and the author of this article had no role in it". Perhaps it would be clearer if this explanation was given at the beginning of the article.
Numbered reactions are sometimes referred to in the text as schemes, and sometimes as reaction equations. Of course, these are reaction equations.
It is not entirely clear on what principle the author of the article put some of figures in the main text, and some in the appendix.
The section 5 "Conclusion" does not look like a typical conclusion of the scientific article. It mainly describes the features of the calculations performed. It would be good to strengthan the biological significance of the research, maybe even moving here a part of the "Discussion" section.
The list of references must be unified. Both names and article titles are sometimes written in capital, sometimes in lowercase lettes. First names are sometimes written in full, sometimes are given with initials, etc. It is necessary to review the entire list, and to correct it according to the journal rules.
And to the same. The reference 49 in the text (line 211) includes also the name and year in square brackets. It should be removed.
Author Response
Dear Reviewer 1, many thanks indeed for kind interest and critical reading. Here appended is my reply to the comments.
Best regards
FMRubino

Reviewer 2 Report
This work does an independent assessment of published results from the glutathione redox system in erythrocytes after challenge with organic oxidants. This work models and extracts the redox potential of the reactive beta B-93-Cysteine of hemoglobin in erythrocytes based on literature results of redox system concentrations. The work assumes the Nernst equation in a closed system and determines the value of -121 mV for the sole reactive cysteine in Hb. Based on this value the work proposes a model for Hb thiol oxidation in the larger context of erythrocyte homeostasis. The work provides a very comprehensive description of the modeling, and provides a good rationale for the process. The work would be a general interest to the community and provides insights into the field. There are some minor points that should be addressed before publication. Specifically, the concentrations of HbSSG presented in different figures seems inconsistent. The authors need to comment on the concentrations, although the rates support their claims, it is not clear if the concentration of mixed HbSSG is sufficient to replenish the GSH equilibrium levels. Figure 1 shows HbSSG levels reaching only 2x10e-5 M whereas the model shows 1.1e-4 M (~2% of 5.5 mM). This is about a 10x difference in concentration and affects the interpretation of the significance of the role Cys-94 plays in redox homeostasis.
Some minor editorial comments:
Line 47: should read met-hemoglobin.
The scheme from lines 116-121 should indicate R’
Lines 199-204: overuse of the unspecific adverbs make it hard to follow.
Figure 5 is hard to follow and should be clarified to improve impact.
Author Response
Dear Reviewer 2, thank you for interest and a much needed critical reading. Here appended is my reply to the comments.
Best regards
FMRubino
